# DemoSG: Demonstration-enhanced Schema-guided Generation for Low-resource Event Extraction

**Gang Zhao, Xiaocheng Gong, Xinjie Yang, Guanting Dong, Shudong Lu** and **Si Li**[*]

School of Artificial Intelligence, Beijing University of Posts and Telecommunications

{zhaogang, xiaochengkung, yangxinjie, lushudong, lisi}@bupt.edu.cn

## Abstract

Most current Event Extraction (EE) methods focus on the high-resource scenario, which requires a large amount of annotated data and can hardly be applied to low-resource domains. To address EE more effectively with limited resources, we propose the Demonstration-enhanced Schema-guided Generation (DemoSG) model, which benefits low-resource EE from two aspects: Firstly, we propose the demonstration-based learning paradigm for EE to fully use the annotated data, which transforms them into demonstrations to illustrate the extraction process and help the model learn effectively. Secondly, we formulate EE as a natural language generation task guided by schema-based prompts, thereby leveraging label semantics and promoting knowledge transfer in low-resource scenarios. We conduct extensive experiments under in-domain and domain adaptation low-resource settings on three datasets, and study the robustness of DemoSG. The results show that DemoSG significantly outperforms current methods in low-resource scenarios.

## 1 Introduction

Event Extraction (EE) aims to extract event records from unstructured texts, which typically include a trigger indicating the occurrence of the event and multiple arguments of pre-defined roles (Doddington et al., 2004a; Ahn, 2006). For instance, the text shown in Figure 1 describes two records corresponding to the *Transport* and *Meet* events, respectively. The *Transport* record is triggered by the word "*arrived*" and consists of 3 arguments:"*Kelly*", "*Beijing*", and "*Seoul*". Similarly, the *Meet* record is triggered by "*brief*" and has two arguments: "*Kelly*" and "*Yoon*". EE plays a crucial role in natural language processing as it provides valuable information for various downstream tasks, including knowledge graph construction (Zhang et al., 2020) and question answering (Han et al., 2021).

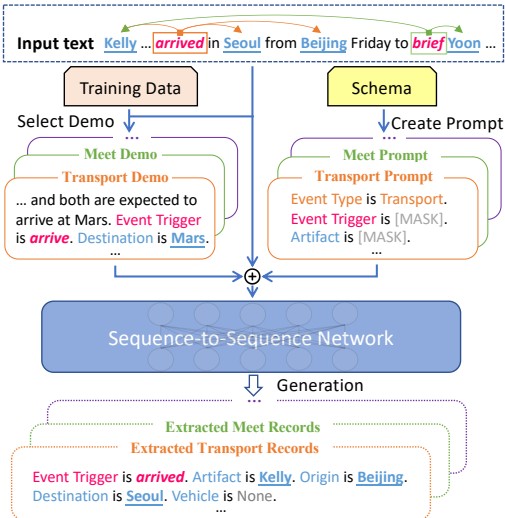

Figure 1: A simplified EE example from the dataset ACE05-E, and the insight of proposed DemoSG. Event triggers, arguments, and roles are highlighted in colors.

Most studies on EE primarily focus on the high-resource scenario (Nguyen et al., 2016; Yan et al., 2019; Cui et al., 2020; Du and Cardie, 2020a; Ramponi et al., 2020; Huang and Peng, 2021), which requires a large amount of annotated training data to attain satisfactory performance. However, event annotation is a costly and labor-intensive process, rendering these methods challenging to apply in domains with limited annotated data. Thus, there is a growing need to explore EE in low-resource scenarios characterized by a scarcity of training examples, which has garnered recent attention.

Lu et al. (2021) models EE as a unified sequence-to-structure generation, facilitating knowledge sharing between the Event Detection and Argument Extraction subtasks. Building upon Lu et al. (2021), Lu et al. (2022) introduces various pre-training strategies on large-scale datasets to enhance structure generation and improve low-resource EE performance. Hsu et al. (2022) incorporates supplementary information by manually designing a prompt for each event type, encompassing event descriptions and role relations. However, despite their effectiveness, existing methods exhibit cer-

---

[*]Corresponding author.

tain inadequacies: 1) Sequence-to-structure Generation ignores the gap between the downstream structure generation and pre-training natural language generation. 2) Large-scale pre-training needs a lot of computational resources and excess corpus. 3) Manual design of delicate prompts still demands considerable human effort, and the performance of EE is sensitive to prompt design.

In this paper, to tackle EE more effectively with limited resources, we propose the **Demo**nstration-enhanced **S**chema-guided **G**eneration (DemoSG) model, benefiting from two aspects: 1) Improving the efficiency of using annotated data. 2) Enhancing the knowledge transfer across different events.

To make full use of annotated training data, we consider them not only as signals for supervising model learning, but also as task demonstrations to illustrate the extraction process to the model. Specifically, as shown in Figure 1, DemoSG selects a suitable training example for each event, transforms it into a demonstration in the natural language style, and incorporates the demonstration along with the input text to enhance the extraction. We refer to this paradigm as demonstration-based learning for EE, which draws inspiration from the in-context learning of the GPT series (Ouyang et al., 2022). To identify appropriate instances for demonstrating, we devise several selecting strategies that can be categorized into two groups: 1) Demo-oriented selection, which aims to choose examples with the best demonstrating features. 2) Instance-oriented retrieval, which aims to find the examples that are most semantically similar to the input sentence. With the enhancement of demonstrations, DemoSG can better understand the EE process and learn effectively with only a few training examples.

To further enhance the knowledge transfer capability and improve the effectiveness of demonstrations, we leverage label semantic information by formulating EE to a seq2seq generation task guided by schema-based prompts. In detail, DemoSG creates a prompt formally similar to the demonstrations for each event, which incorporates the event type and roles specified in the event schema. Then, the prompt and demo-enhanced sentence are fed together into the pre-trained language model (PLM) to generate a natural sentence describing the event records. Finally, DemoSG employs a rule-based decoding algorithm to decode records from the generated sentences. The schema-guided sequence generation of DemoSG promotes knowledge transfer in

multiple ways: Firstly, leveraging label semantics facilitates the extraction and knowledge transfer across different events. For instance, the semantic of the "*Destination*" role hints that its argument is typically a place, and the similarity between "*Attacker*" and "Adjudicator" makes it easy to transfer argument extraction knowledge among them, for they both indicate to humans. Secondly, natural language generation is consistent with the pre-training task of PLM, eliminating the need for excessive pre-training of structural generation methods (Lu et al., 2021, 2022). Furthermore, unlike classification methods constrained by predefined categories, DemoSG is more flexible and can readily adapt to new events through their demonstrations without further fine-tuning, which is referred to as the parameter-agnostic domain adaptation capability.

Our contributions can be summarized as follows:

1) We propose a demonstration-based learning paradigm for EE, which automatically creates demonstrations to help understand EE process and learn effectively with only a few training examples.

2) We formulate EE to a schema-guided natural language generation, which leverages the semantic information of event labels and promotes knowledge transfer in low-resource scenarios.

3) We conduct extensive experiments under various low-resource settings on three datasets and study the robustness of DemoSG. The results show that DemoSG significantly outperforms previous methods in low-resource scenarios.

## 2 Methodology

In this section, we first elucidate some preliminary concepts of low-resource EE in Section 2.1. Next, we present the proposed Demonstration-enhanced Schema-guided Generation model in Section 2.2. Lastly, we provide details regarding the training and inference processes in Section 2.3.

### 2.1 Low-resource Event Extraction

Given a tokenized input sentence $X = \{x_i\}_{i=1}^{|X|}$, the event extraction task aims to extract a set of event records $\mathbf{R} = \{R_j\}_{j=1}^{|\mathbf{R}|}$, where $R_j$ contains a trigger $T_j$ and several arguments $\mathbf{A}_j = \{A_{jk}\}_{k=1}^{|\mathbf{A}_j|}$. Each $T_j$ or $A_{jk}$ corresponds to an event type $E_j$ or event role $O_{jk}$ predefined in the schema $\mathcal{S}$. This paper considers two types of low-resource EE scenarios:

**In-domain low-resource scenario** focuses on the challenge that the amount of training examples is quite limited. Considering the complete train-

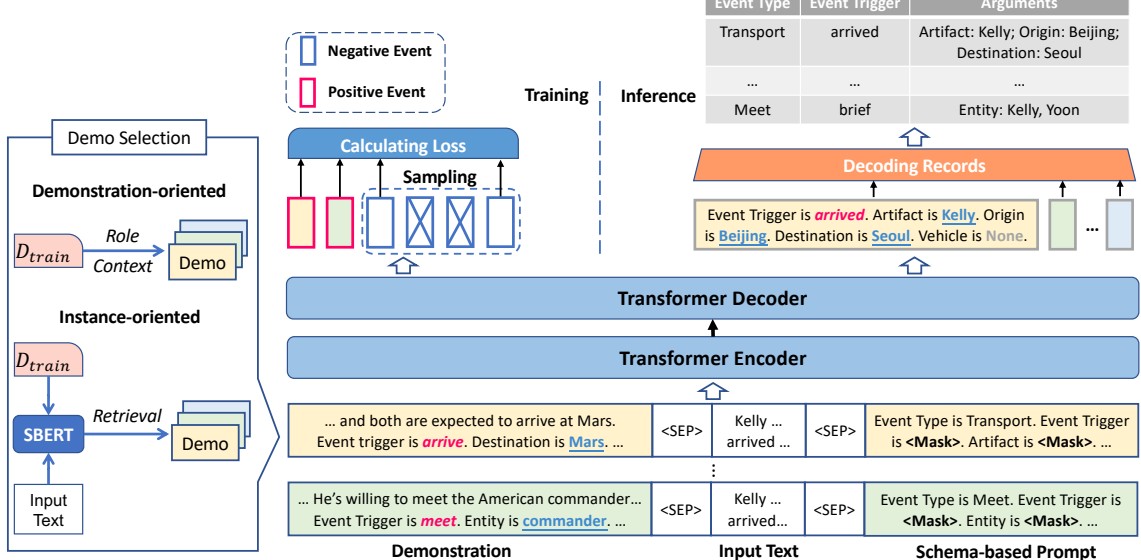

Figure 2: The overall architecture of DemoSG. The left shows the demonstration selection process, while the right presents the demonstration-enhanced record generation framework. Different events are distinguished using colors.

ing dataset $\mathcal{D}_h = \{(X_i, \mathbf{R}_i)\}_{i=1}^{|\mathcal{D}_h|}$ with the schema $\mathcal{S}_h$ and a subset $\mathcal{D}_l$ with $\mathcal{S}_l \subset \mathcal{S}_h$, the objective is to fully use the limited data and train a high-performance model on $\mathcal{D}_l$ when $|\mathcal{D}_l| \ll |\mathcal{D}_h|$. The in-domain low-resource scenario poses challenges to the data utilization efficiency of EE models.

**Domain adaptation scenario** pertains to the situation where the target domain lacks examples but a large number of source domain examples are available. Given two subsets $\mathcal{D}_{src}$ and $\mathcal{D}_{tgt}$, where $\mathcal{S}_{src} \cap \mathcal{S}_{tgt} = \oslash$ and $|\mathcal{D}_{tgt}| \ll |\mathcal{D}_{src}|$, our objective is to initially pre-train a source-domain model on $\mathcal{D}_{src}$, then achieve high performance on target domain subset $\mathcal{D}_{tgt}$. Domain adaptation enables the low-resource domain to leverage the knowledge acquired from the well-studied domain, requiring a strong knowledge transfer capability of EE models.

## 2.2 Demonstration-enhanced Schema-guided Generation for Event Extraction

We propose an end-to-end model DemoSG, which leverages demonstration-based learning and schema-guided generation to address the aforementioned low-resource scenarios. As Figure 2 shows, DemoSG initially constructs the task demonstration and prompt for each event type, and then employs a sequence-to-sequence network to generate natural sentences that describe the event records.

### 2.2.1 Unified Sequence Representation of Event Record

To model EE as a sequence-to-sequence generation task, we design a unified representation template to transform event records to unambiguous natural

sentences that include event triggers, arguments, and their corresponding event roles.

In detail, given a record $\mathbf{R}$ with a trigger $T$ and several arguments $\mathbf{A} = \{A_i\}_{i=1}^{|\mathbf{A}|}$, where each $A_i$ corresponds to the role $O_i$, DemoSG transfers $\mathbf{R}$ to a sequence $Y = \{y_j\}_{j=1}^{|Y|}$ that is designed as "*Event trigger is $T$ . $O_1$ is $A_1$ . $O_2$ is $A_2$...*". Those roles without any arguments are followed by the padding token "*None*". For example, the *Transport* record in Figure 2 is represented as "*Event trigger is arrived. Artifact is Kelly. Origin is Beijing. Destination is Seoul. Vehicle is None...*". We also consider the following special situations: Multiple records of the same event type can be expressed by concatenating respective sequence representations. Multiple arguments corresponding to the same role will be merged together via "*&*", such as "*Artifact is Kelly & Yoon*". Since DemoSG extracts each type of events separately, event type is no longer required for record representations, which relieves the pressure of model prediction.

### 2.2.2 Event Demonstration Construction

To effectively use the limited training examples, we not only treat them as traditional supervised learning signal, but also transform them to event demonstrations which can bring additional information and help understand the extraction process.

The demonstration $D_i$ of event type $E_i$ is a natural sentence containing a context part and an annotation part, which is constructed by the following steps: Firstly, DemoSG selects or retrieves an example $(X_i, \mathbf{R}_i)$ from the training set $\mathcal{D}_{train}$, which

contains records of $E_i$. Next, DemoSG transforms records associated with $E_i$ to an annotation sentence $Y_i$ following the record representation template in Section 2.2.1. Note that we employ the unified record template for both demonstrations and model predictions, promoting the cohesive interaction between them. Finally, the demonstration $D_i$ is constructed by concatenating the context part $X_i$ and the annotation part $Y_i$. Given the significance of selecting appropriate examples for constructing demonstrations, we propose several selection strategies that can be categorized into two groups:

**Demonstration-oriented selection** aims to pick the examples with the best demonstrating characteristics. Specifically, the training sample associated with more event roles tends to contain more information of extracting such type of events. And an example with longer text may offer more contextual information for extracting the same event. Based on these considerations, we propose two selection strategies: 1) *rich-role* strategy selects the example with the highest number of associated roles for each event. 2) *rich-context* strategy chooses the example with the longest context for each event.

**Instance-oriented retrieval** focuses on retrieving examples that are most semantically *similar* to the input sentence, as the semantic similarity may enhance the effectiveness of demonstrations. Concretely, *similar* strategy involves encoding the input sentence $X$ and each example sentence $X_i$ using SBERT (Reimers and Gurevych, 2019), followed by calculating the cosine similarity between their *[CLS]* embeddings to rank $X_i$. Finally, *similar* strategy retrieves the top-ranked example for each event type to construct its demonstration.

### 2.2.3 Schema-based Prompt Construction

We design a prompt template for DemoSG to exploit the semantic information of event types and roles base on the event schema. Given the event schema $\mathcal{S} = \{E_i, \mathbf{O}_i\}_{i=1}^{N_E}$, where $E_i$ is the event type and $\mathbf{O}_i = \{O_{ij}\}_{j=1}^{|\mathbf{O}_i|}$ are event roles, the prompt of $E_i$ is designed as: "*Event type is $E_i$. Event trigger is <Mask>. $O_{i1}$ is <Mask>. $O_{i2}$ is <Mask>...*", where *<Mask>* represent the mask token of the PLM. For example, the prompt of aforementioned *Transport* event can be constructed as "*Event type is Transport. Event trigger is <Mask>. Artifact is <Mask>. Origin is <Mask>...*". Leveraging the label semantic information not only helps query relevant triggers and arguments, but also facilitates knowledge transfer across different events.

### 2.2.4 Enhanced Sequence-to-sequence Generation for Event Extraction

As Figure 2 shows, DemoSG generates the record sequence of each event type individually via a common encoder-decoder architecture enhanced by respective event demonstration and prompt. Given an input sentence $X = \{x_j\}_{j=1}^{|X|}$, DemoSG first constructs event demonstration $D_i = \{d_{ij}\}_{j=1}^{|D_i|}$ and schema-based prompt $P_i = \{p_{ij}\}_{j=1}^{|P_i|}$ for the event type $E_i$. Then, DemoSG concatenates $D_i$, $X$ and $P_i$ via the "*<SEP>*" token, and uses a Transformer Encoder (Vaswani et al., 2017) to obtain the demonstration-enhanced hidden representation:

$$\mathbf{H}_i = Encoder([D_i; X; P_i]) \qquad (1)$$

Subsequently, DemoSG decodes the enhanced representation $\mathbf{H}_i$ and generates event record sequence $Y_i = \{y_{ij}\}_{j=1}^{|Y_i|}$ token by token:

$$y_{ij}, \mathbf{h}_{ij} = Decoder([\mathbf{H}_i; \mathbf{h}_{i1}, ..., \mathbf{h}_{i,j-1}]) \qquad (2)$$

where $Decoder(\cdot)$ represents the transformer decoder and $\mathbf{h}_{ij}$ is the decoder state at the $j^{th}$ step. By iterating above generation process for all event types $\{E_i\}_{i=1}^{N_E}$, DemoSG finally obtains the complete record sequence set $\mathbf{Y} = \{Y_i\}_{i=1}^{N_E}$.

### 2.3 Training and Inference

Since DemoSG generates records for each event type individually, we consider the sentences generated for annotated event types as positive examples, while the sentences generated for unannotated event types serve as negative examples. At the training stage, we sample negative examples $m$ times the number of positive samples, where $m$ is a hyperparameter. The following negative log-likelihood loss function is optimized when training:

$$\mathcal{L} = - \sum_{\mathcal{D}_P \cup \mathcal{D}_N} \log p(Y|X, \mathcal{D}_{train}, \mathcal{S}, \theta) \qquad (3)$$

$$p(Y|X, \mathcal{D}_{train}, \mathcal{S}) = \prod_{i=1}^{|Y|} p(y_i|y_{<i}, \mathcal{D}_{train}, \mathcal{S}) \qquad (4)$$

where $\theta$ represents the model parameters, $\mathcal{D}_{train}$ is the training set, $\mathcal{S}$ is the event schema, $\mathcal{D}_P$ and $\mathcal{D}_N$ are the positive and sampled negative sets.

At the inference phase, DemoSG decodes event records from the generated $\mathbf{Y} = \{Y_i\}_{i=1}^{N_E}$ via a rule-based deterministic algorithm, and employs string matching to obtain the offsets of event triggers and arguments. Following Lu et al. (2021), when the predicted string appears multiple times in the sentence, we choose all matched offsets for trigger prediction, and the matched offset that is closest to the predicted trigger for argument extraction.

| Dataset | Statistic | Dev | Test | Train | | | | | | |
|---------|-----------|-----|------|-------|-----|-----|-----|-------|-------|--------|
| | | | | Full(100%) | 2% | 5% | 10% | 2shot | 5shot | 10shot |
| ACE05-EN | #Sents | 923 | 832 | 17,172 | 59 | 154 | 314 | 66 | 157 | 302 |
| | #Events | 450 | 403 | 4,202 | 74 | 206 | 414 | 101 | 234 | 458 |
| | #Roles | 605 | 576 | 4,859 | 84 | 225 | 442 | 106 | 247 | 487 |
| ACE05-EN+ | #Sents | 901 | 676 | 19,216 | 71 | 176 | 348 | 66 | 158 | 300 |
| | #Events | 468 | 424 | 4,419 | 91 | 232 | 455 | 100 | 241 | 448 |
| | #Roles | 759 | 689 | 6,607 | 114 | 323 | 609 | 140 | 325 | 605 |

Table 1: Data statistics of ACE05-EN and ACE05-EN+. #Sents, #Events and #Roles indicate the sentence, event record and role numbers. Statistics of the low-resource subsets are the average results of five different sampling.

# 3 Experiments

To assess the effectiveness of DemoSG, we conduct comprehensive experiments under in-domain low-resource, domain adaptation, and high-resource settings. An ablation study is applied to explore the impact of each module and the improvement achieved. Furthermore, we also investigate the robustness of applying demonstrations of DemoSG.

## 3.1 Experimental Settings

**Datasets.** We evaluate our method on two widely used event extraction benchmarks: **ACE05-EN** (Wadden et al., 2019) and **ACE05-EN$^+$** (Lin et al., 2020). Both of them contain 33 event types as well as 22 event roles, and derive from **ACE2005** (Doddington et al., 2004b)[1], a dataset that provides rich annotations of entities, relations and events in English. The full data splits and preprocessing steps for both benchmarks are consistent with previous works (Wadden et al., 2019; Lin et al., 2020). Additionally, we utilize the same data sampling strategy as UIE (Lu et al., 2022) for the low-resource settings. Detail statistics of the datasets are shown in Table 1.

**Evaluation Metrics.** We utilize the same evaluation metrics as previous event extraction studies (Wadden et al., 2019; Lin et al., 2020; Lu et al., 2021, 2022): 1) Trigger Classification Micro F1-score (**Trig-C**): a trigger is correctly classified if its offset and event type align with the gold labels. 2) Argument Classification Micro F1-score (**Arg-C**): an argument is correctly classified if its event type, offset, and event role all match the golden ones.

**Baselines.** We compare our method with the following baselines in low-resource scenarios: 1) **OneIE** (Lin et al., 2020), the current SOTA high-resource method, which extracts globally optimal event records using global features. 2) **Text2Event** (Lu et al., 2021), which integrates

event detection and argument extraction into a unified structure generation task. 3) **UIE** (Lu et al., 2022), which improves the low-resource performance through various pre-training strategies based on Text2Event. 4) **DEGREE** (Hsu et al., 2022), which incorporates additional information by manually designing a prompt for each event type.

For the high-resource experiments, we also compare with the following methods: 1) **DY-GIE++** (Wadden et al., 2019), which is a BERT-based classification model that utilizes the span graph propagation. 2) **Joint3EE** (Nguyen and Nguyen, 2019), which jointly extracts entities, triggers, and arguments based on the shared representations. 3) **GAIL** (Zhang et al., 2019), which is a joint entity and event extraction model based on inverse reinforcement learning. 4) **EEQA** (Du and Cardie, 2020b) and **MQAEE** (Li et al., 2020), which formulate event extraction as a question answering problem using machine reading comprehension models. 5) **TANL** (Paolini et al., 2021), which formulates event extraction as a translation task between augmented natural languages.

**Implementations.** We employ the BART-large pre-trained by Lewis et al. (2020) as our seq2seq network, and optimize our model via the Adafactor (Shazeer and Stern, 2018) optimizer with a learning rate of 4e-5 and a warmup rate of 0.1. The training batch size is set to 16, and the epoch number for the in-domain low-resource experiments is set to 90, while 45 for the domain adaptation and high-resource experiments of DemoSG. Since ONEIE is not specifically designed for low-resource scenarios, we train it for 90 epochs as well in the in-domain setting. The negative example sampling rate $m$ is set to 11 after the search from {5,7,9,11,13,15} using dev set on the high-resource setting, and we find that $m$ can balance Recall and Precision scores in practice. We conduct experiments on the single Nvidia Tesla-V100 GPU. All experiments of baseline methods are conducted based on the released code of their orig-

---

[1] https://catalog.ldc.upenn.edu/LDC2006T06

| Metric | Models | ACE05-EN | | | | | | | | ACE05-EN+ | | | | | | | |
|---|---|---|---|---|---|---|---|---|---|---|---|---|---|---|---|---|---|
| | | 2-shot | 5-shot | 10-shot | AVE-S | 2% | 5% | 10% | AVE-R | 2-shot | 5-shot | 10-shot | AVE-S | 2% | 5% | 10% | AVE-R |
| **Trig-C F1(%)** | OneIE | $37.3_{\pm3.4}$ | $55.4_{\pm2.3}$ | **$61.9_{\pm4.9}$** | 51.5 | $34.2_{\pm5.5}$ | $51.6_{\pm4.5}$ | $\underline{61.3_{\pm1.3}}$ | 49.0 | $31.1_{\pm2.6}$ | $56.0_{\pm0.6}$ | $61.2_{\pm1.4}$ | 49.4 | $40.7_{\pm4.2}$ | $55.4_{\pm5.2}$ | $65.3_{\pm2.2}$ | 53.8 |
| | Text2Event | $20.2_{\pm2.0}$ | $29.6_{\pm0.3}$ | $33.6_{\pm2.0}$ | 27.8 | $35.0_{\pm2.9}$ | $49.0_{\pm3.4}$ | $58.0_{\pm0.1}$ | 47.3 | $25.0_{\pm1.3}$ | $32.6_{\pm1.6}$ | $39.8_{\pm0.3}$ | 32.5 | $39.5_{\pm2.4}$ | $49.0_{\pm4.1}$ | $58.0_{\pm1.6}$ | 48.8 |
| | DEGREE | $27.9_{\pm3.2}$ | $32.0_{\pm1.1}$ | $46.3_{\pm1.2}$ | 35.4 | $43.7_{\pm0.2}$ | $45.8_{\pm1.2}$ | $51.3_{\pm2.4}$ | 46.9 | $26.5_{\pm3.5}$ | $32.6_{\pm1.7}$ | $42.5_{\pm0.2}$ | 33.9 | $43.1_{\pm3.3}$ | $41.0_{\pm0.5}$ | $45.1_{\pm2.5}$ | 43.0 |
| | UIE | $42.6_{\pm2.5}$ | $48.3_{\pm1.1}$ | $54.1_{\pm0.6}$ | 48.3 | **$50.1_{\pm0.4}$** | $57.3_{\pm0.9}$ | $60.1_{\pm0.9}$ | **55.8** | $40.4_{\pm7.1}$ | $48.6_{\pm4.6}$ | $51.9_{\pm1.5}$ | 47.0 | **$52.8_{\pm1.2}$** | $55.2_{\pm2.0}$ | $60.1_{\pm2.3}$ | 56.0 |
| | DemoSG$_R$ | $49.7_{\pm3.4}$ | $\underline{59.3_{\pm1.2}}$ | $\underline{60.1_{\pm1.1}}$ | $\underline{56.4}$ | $46.6_{\pm5.8}$ | **$59.0_{\pm3.2}$** | $60.4_{\pm2.2}$ | $\underline{55.3}$ | $45.7_{\pm3.5}$ | $55.6_{\pm2.1}$ | $\underline{63.1_{\pm2.6}}$ | 54.8 | $46.6_{\pm8.7}$ | $\underline{60.5_{\pm1.7}}$ | **$63.3_{\pm0.4}$** | $\underline{56.8}$ |
| | DemoSG$_C$ | $\underline{51.4_{\pm2.3}}$ | **$59.7_{\pm2.0}$** | $58.8_{\pm0.9}$ | **56.6** | $46.6_{\pm4.8}$ | $\underline{57.6_{\pm4.1}}$ | **$61.7_{\pm0.9}$** | $\underline{55.3}$ | $\underline{48.6_{\pm5.3}}$ | $58.2_{\pm4.3}$ | **$64.0_{\pm0.7}$** | **56.9** | $46.8_{\pm4.5}$ | **$61.3_{\pm1.6}$** | $\underline{63.0_{\pm0.9}}$ | **57.0** |
| | DemoSG$_S$ | **$51.7_{\pm5.2}$** | $57.0_{\pm1.69}$ | $59.9_{\pm2.5}$ | 56.2 | $\underline{47.3_{\pm4.5}}$ | $53.6_{\pm9.3}$ | $60.0_{\pm2.3}$ | 53.6 | **$50.9_{\pm6.1}$** | $\underline{57.5_{\pm1.4}}$ | $61.4_{\pm2.6}$ | $\underline{56.6}$ | $\underline{49.5_{\pm4.6}}$ | $58.6_{\pm0.6}$ | $60.5_{\pm2.4}$ | 56.2 |
| **Arg-C F1(%)** | OneIE | $4.6_{\pm0.3}$ | $14.2_{\pm1.9}$ | $24.2_{\pm3.4}$ | 14.3 | $7.3_{\pm2.0}$ | $18.4_{\pm1.4}$ | $29.6_{\pm1.9}$ | 18.4 | $5.6_{\pm0.7}$ | $15.2_{\pm1.1}$ | $24.2_{\pm2.0}$ | 15.0 | $7.9_{\pm1.4}$ | $22.4_{\pm3.2}$ | $33.6_{\pm4.3}$ | 21.3 |
| | Text2Event | $11.1_{\pm1.5}$ | $18.3_{\pm2.2}$ | $22.5_{\pm1.2}$ | 17.3 | $14.3_{\pm2.5}$ | $24.4_{\pm0.2}$ | $32.9_{\pm3.0}$ | 23.9 | $14.1_{\pm1.4}$ | $20.7_{\pm1.6}$ | $26.5_{\pm0.6}$ | 20.4 | $17.4_{\pm1.5}$ | $28.3_{\pm0.9}$ | $35.1_{\pm2.4}$ | 26.9 |
| | DEGREE | $9.8_{\pm1.0}$ | $19.5_{\pm1.8}$ | $25.0_{\pm1.2}$ | 18.1 | $16.7_{\pm6.7}$ | $21.6_{\pm1.9}$ | $28.2_{\pm1.0}$ | 22.2 | $12.1_{\pm0.9}$ | $18.2_{\pm0.9}$ | $27.0_{\pm0.4}$ | 19.1 | $18.8_{\pm1.8}$ | $28.1_{\pm2.3}$ | $31.4_{\pm1.5}$ | 26.1 |
| | UIE | $18.1_{\pm0.9}$ | $25.8_{\pm1.7}$ | $31.6_{\pm1.1}$ | 25.2 | $\underline{21.1_{\pm0.7}}$ | $28.2_{\pm3.4}$ | $33.1_{\pm0.7}$ | 27.5 | $19.3_{\pm1.6}$ | $27.9_{\pm1.5}$ | $30.6_{\pm1.1}$ | 25.9 | $\underline{24.4_{\pm1.6}}$ | $29.1_{\pm1.6}$ | $36.0_{\pm2.0}$ | 29.8 |
| | DemoSG$_R$ | $16.6_{\pm1.1}$ | $\underline{34.8_{\pm2.0}}$ | $\underline{38.3_{\pm1.0}}$ | 29.9 | $18.3_{\pm2.2}$ | $\underline{32.3_{\pm2.9}}$ | **$41.3_{\pm2.5}$** | $\underline{30.6}$ | $22.3_{\pm1.8}$ | $\underline{34.4_{\pm2.9}}$ | $42.0_{\pm3.1}$ | 32.9 | $22.0_{\pm5.0}$ | $\underline{38.0_{\pm1.6}}$ | $42.2_{\pm1.6}$ | 34.1 |
| | DemoSG$_C$ | $\underline{22.5_{\pm3.2}}$ | **$35.8_{\pm1.4}$** | $36.4_{\pm2.4}$ | $\underline{31.6}$ | $15.8_{\pm3.3}$ | $31.2_{\pm5.7}$ | $\underline{37.2_{\pm2.5}}$ | 28.1 | $\underline{23.5_{\pm4.5}}$ | $34.4_{\pm2.8}$ | $41.7_{\pm0.3}$ | $\underline{33.2}$ | $24.2_{\pm1.8}$ | **$38.8_{\pm0.9}$** | **$42.5_{\pm2.2}$** | $\underline{35.2}$ |
| | DemoSG$_S$ | **$25.5_{\pm3.8}$** | $33.3_{\pm0.8}$ | **$39.1_{\pm3.1}$** | **32.6** | **$22.2_{\pm3.2}$** | **$34.7_{\pm2.0}$** | $40.0_{\pm1.3}$ | **32.3** | **$28.7_{\pm4.8}$** | **$35.6_{\pm4.9}$** | $40.5_{\pm1.8}$ | **34.9** | **$28.3_{\pm2.1}$** | $37.1_{\pm1.0}$ | **$42.5_{\pm2.4}$** | **36.0** |

Table 2: Experimental results in the in-domain low-resource settings. AVE-S (hot) and AVE-R (atio) are average performances across the few-shot and the data-limited settings, respectively. DemoSG$_{R(ole)}$, DemoSG$_{C(ontext)}$ and DemoSG$_{S(imilar)}$ represent three variants of DemoSG with different demonstration selecting strategies. We report the means and standard deviations of 5 sampling seeds to mitigate the randomness introduced by data sampling.

inal papers. Considering the influence of sampling training examples in low-resource experiments, we run 5 times with different sampling seeds and report the average results as well as standard deviations. Our code will be available at https://github.com/GangZhao98/DemoSG.

## 3.2 In-domain Low-resource Scenario

To verify the effectiveness of DemoSG in in-domain low-resource scenarios, we conduct extensive experiments in several *few-shot* and *data-limited* settings, following Lu et al. (2022). For the few-shot experiments, we randomly sample 2/5/10 examples per event type from the original training set, while keeping the dev and test sets unchanged. For the data-limited experiments, we directly sample 2%/5%/10% from the original training set for model training. Compared with the few-shot setting, the data-limited setting has unbalanced data distributions and zero-shot situations, posing challenges to the generalization ability of EE methods.

Table 2 presents the results of in-domain low-resource experiments. We can observe that:

1) DemoSG shows remarkable superiority over other baselines in the in-domain low-resource scenarios. For argument extraction, DemoSG consistently achieves improvements on few-shot and data-limited settings compared to baseline methods. For instance, in the 2/5/10-shot settings of ACE05-EN and ACE05-EN+, DemoSG outperforms the highest baseline methods by +7.4%/+10.0%/+7.5% and +9.4%/+7.7%/+11.4% in terms of Arg-C F1, respectively. Regarding event detection, DemoSG also surpasses the highest baselines by +1.0% on AVE-R of ACE05-EN+, and +5.1%/+7.5% on

AVE-S of two datasets. These results provide compelling evidence for the effectiveness of our method in low-resource event extraction.

2) It is noteworthy that DemoSG exhibits greater improvement in few-shot settings compared to data-limited settings, and the extent of improvement in data-limited settings increases with the growth of available data. Specifically, for argument extraction, DemoSG achieves a +4.8%/+6.2% improvement in AVE-R, whereas it achieves a higher improvement of +7.4%/+9.0% in AVE-S on the two benchmarks. Furthermore, in the data-limited settings of the two benchmarks, DemoSG demonstrates a +1.1%/+6.5%/+8.2% and +3.9%/+9.7%/+6.5% increase in Arg-C F1, respectively. These observations suggest that our proposed demonstration-based learning may be more effective when there is a more balanced data distribution or greater availability of demonstrations.

3) All of the proposed demonstration selection strategies achieve strong performance and possess distinct characteristics when compared to each other. The *similar* retrieving strategy excels in tackling low-resource argument extraction, with a 2.7%/2.0% higher AVE-S of Arg-C than *rich-role* on the two benchmarks. And the *rich-context* strategy tends to have better low-resource event detection ability, with a 1.7%/0.8% higher AVE-R of Trig-C than *similar* retriving on both benchmarks.

## 3.3 Domain Adaptation Scenario

To investigate the cross-domain knowledge transfer capability of DemoSG, we perform experiments in both the *parameter-adaptive* and *parameter-agnostic* domain adaptation settings.

| Model | ACE05-EN | | | | ACE05-EN+ | | | |
|---|---|---|---|---|---|---|---|---|
| | Parameter-adaptive | | Parameter-agnostic | | Parameter-adaptive | | Parameter-agnostic | |
| | Trig-C | Arg-C | Trig-C | Arg-C | Trig-C | Arg-C | Trig-C | Arg-C |
| OneIE | **83.1** | 58.8 | - | - | 77.2 | 50.9 | - | - |
| Text2Event | 77.3 | 61.6 | 12.1 | 9.7 | 79.8 | 59.7 | 22.1 | 20.3 |
| DEGREE | 81.0 | 63.4 | 26.9 | 18.1 | 81.2 | 61.4 | 33.9 | 26.3 |
| UIE | 81.4 | 62.9 | 35.2 | 11.2 | 81.3 | 61.7 | 41.2 | 22.5 |
| DemoSG$_R$ | 82.4$^\dagger$ | **69.2** $^\dagger$ | **45.9**$^\dagger$ | **36.9**$^\dagger$ | **81.4**$^\dagger$ | **63.7**$^\dagger$ | **44.3**$^\dagger$ | **34.9**$^\dagger$ |

Table 3: Performance comparison of Trig-C and Arg-C F1 scores(%) in *parameter-adaptive* and *parameter-agnostic* domain adaptation settings. Given that OneIE is classification-based, adapting it to target domains without fine-tuning poses a significant challenge. The marker † refers to significant test *p-value < 0.05* when comparing with DEGREE.

| Model | Type | ACE05-EN | | ACE05-EN+ | |
|---|---|---|---|---|---|
| | | Trig-C | Arg-C | Trig-C | Arg-C |
| DYGIE++* | Cls | 73.6 | 52.5 | - | - |
| Joint3EE* | Cls | 69.8 | 52.1 | - | - |
| GAIL* | Cls | 72.0 | 52.4 | - | - |
| EEQA* | Cls | 72.4 | 53.3 | - | - |
| MQAEE* | Cls | 71.7 | 53.4 | - | - |
| TANL* | Gen | 68.4 | 47.6 | - | - |
| OneIE | Cls | **74.2** | **57.0** | **72.5** | 55.9 |
| Text2Event | Gen | 71.9* | 53.8* | 70.3 | 53.4 |
| DEGREE† | Gen | 73.3* | 55.8* | 70.7 | 55.7 |
| UIE | Gen | 73.4* | 54.8* | 70.7 | 52.6 |
| DemoSG$_R$ | Gen | 73.4$^\dagger$ | 56.0$^\dagger$ | 71.2$^\dagger$ | **56.8** $^\dagger$ |

Table 4: Performance comparison in the high-resource setting. "Cls" and "Gen" stand for classification-based and generation-based method. Results marked with * are sourced from the original paper, and † denotes significant test *p-value < 0.05* compared with DEGREE.

**Parameter-adaptive domain adaptation.** Following Lu et al. (2021), in the parameter-adaptive setting, we divide each dataset into a source domain subset *src* and a target domain subset *tgt*. The *src* keeps the examples associated with the top 10 frequent event types, and the *tgt* keeps the sentences associated with the remaining 23 event types. For both *src* and *tgt*, we sample 80% examples for model training and use the other 20% for evaluation. After the sampling, we first pre-train a source domain model on the *src*, and then fine-tune the model parameters and evaluate on the *tgt* set. As shown in Table 3, DemoSG outperforms the baseline methods in argument extraction for the target domain, exhibiting a +5.8%/+2.0% improvement in Arg-C F1 on ACE05-EN/ACE05-EN+ compared to the best-performing baselines. For event detection, DemoSG also achieves the highest performance on ACE05-EN+ and performs competitively with the SOTA method ONEIE on ACE05-EN. The results suggest that DemoSG possesses strong domain adaptation capabilities by leveraging the label semantic information of the event schema.

**Parameter-agnostic domain adaptation.** Unlike previous event extraction methods, we enhance the generation paradigm via demonstration-based learning, enabling DemoSG to adapt to new domains and event types without further finetuning model parameters on the target domain. Specifically, DemoSG can directly comprehend the extraction process of new event types via demonstrations of the target domain at the inference phase. We regard this ability as the *parameter-agnostic* domain adaptation, which can avoid the catastrophic forgetting (Li et al., 2022) and the extra computing cost brought by the finetuning process. For the parameter-agnostic setting, we first train a source domain model on the *src*, and then directly evaluate on the *tgt* set without parameter finetuning. As Table 3 shows, DemoSG gains a significant improvement in both event detection and argument extraction on both datasets. Regarding event detection, DemoSG surpasses the highest baseline UIE by +10.7%/+3.1% on Trig-C F1, benefiting from its ability to comprehend target domain extraction through demonstrations. In terms of argument extraction, although DEGREE showcases strong performance in the parameter-agnostic setting by incorporating role relation information of new event types via manually designed prompts, DemoSG still outperforms DEGREE on both datasets by +18.8%/+8.6% on Arg-C F1. This outcome not only validates the effectiveness of demonstrations for DemoSG but also suggests that demonstration-based learning is a more effective approach to help comprehend the task than intricate prompts.

## 3.4 High-resource Scenario

To gain insights into our framework, we also evaluate our method in the high-resource scenario, where each type of training example is abundant. For the high-resource experiments, we train all models on the full training sets and evaluate their performance on the original development and test sets.

According to Table 4, although designed for low-

| Model | ACE05-EN | | | | | | ACE05-EN+ | | | | | |
|---|---|---|---|---|---|---|---|---|---|---|---|---|
| | 5-shot | | Parameter-adaptive | | High-resource | | 5-shot | | Parameter-adaptive | | High-resource | |
| | Trig-C | Arg-C | Trig-C | Arg-C | Trig-C | Arg-C | Trig-C | Arg-C | Trig-C | Arg-C | Trig-C | Arg-C |
| **DemoSG**$_R$ | 59.3 | 34.8 | 82.4 | 69.2 | 73.4 | 56.0 | 55.6 | 34.4 | 81.4 | 63.7 | 71.2 | 56.8 |
| w/o Demo | -3.7 | -5.5 | -3.0 | -0.4 | -5.4 | -3.5 | -2.8 | -3.2 | -2.8 | -0.8 | -4.1 | -1.3 |
| w/o Schema | -3.9 | -5.9 | -4.9 | -5.4 | -6.9 | -5.3 | -3.6 | -4.3 | -4.5 | -1.9 | -6.8 | -3.8 |
| w/o Demo & Schema | -39.0 | -26.5 | -20.0 | -18.4 | -26.8 | -20.2 | -35.9 | -21.5 | -36.7 | -20.2 | -19.6 | -13.8 |

Table 5: Ablation study for the key components of DemoSG$_{R(ole)}$ under the 5-shot, parameter-adaptive and the high-resource settings. Trig-C and Arg-C represent Trig-C F1 score and Arg-C F1 score, respectively.

resource event extraction, our DemoSG also outperforms baselines by a certain margin in the argument extraction (Arg-C) in the high-resource setting on ACE05-EN+. In terms of the event detection (Trig-C), DemoSG also achieves competitive results on both benchmarks (73.4% and 71.2%). The above results prove that DemoSG has good generalization ability in both low-resource and high-resource scenarios. Furthermore, we observe that generative methods perform better than classification-based methods in many cases under both the low-resource and high-resource scenarios, which illustrates the correctness and great potential of our choice to adopt the generative event extraction framework.

### 3.5 Ablation Study

To examine the impact of each module of DemoSG and the resulting improvement, we perform ablation experiments on three variants of DemoSG in the few-shot, parameter-adaptive, and high-resource settings: 1) *w/o Demo*, which eliminates demonstrations and generates records solely based on concatenated input text and schema-based prompts. 2) *w/o Schema*, which excludes the utilization of schema semantics by replacing all labels with irrelevant tokens during prompt construction. 3) *w/o Demo&Schema*, which removes both demonstrations and schema semantics. From Table 5 we can observe that:

1) The performance of DemoSG experiences significant drops when removing demonstrations in three different scenarios, particularly -5.5%/-3.2% for Arg-C F1 in the 5-shot setting on ACE05-EN/ACE05-EN+. This result indicates that demonstration-based learning plays a crucial role in our framework, especially for in-domain low-resource event extraction.

2) Incorporating label semantic information of the event schema is essential for DemoSG, as it significantly influences all settings. For instance, removing the schema semantic information results in a noteworthy -5.4/-1.9 decrease in Arg-C F1 for the parameter-adaptive domain adaptation setting.

3) We observe that removing both demonstra-

tions and the schema semantic information leads to a substantial performance degradation compared to removing only one of them, particularly -26.5%/-21.5% for Arg-C F1 in the 5-shot setting.

## 4 Effectiveness and Robustness of Demonstration-based Learning

Since demonstrations can influence the low-resource EE performance, we design two variants for DemoSG with different kinds of damage to the demonstrations to explore the effectiveness and robustness of demonstration-based learning:

1) *Demonstration Perturbation.* To analyze the influence of incorrect demonstrations, we sample 40% of the demonstrations and replace the golden triggers and arguments with random spans within the context. *Test Perturbation* applies perturbation during the inference phase only, while *Train-test Perturbation* applies perturbation during both the training and inference phases.

2) *Demonstration Dropping.* To investigate the robustness of demonstration-based learning to missing demonstrations, we randomly drop 40% of the demonstrations during the training or inference phases. *Test Drop* performs dropping during the inference phase only, while *Train-test Drop* applies dropping during both training and inference.

We conduct the above experiments based on DemoSG with *rich-role* strategy under the 5-shot setting. From Figure 3, we can observe that:

1) Incorrect demonstrations exert a detrimental influence on the model performance. Specifically, perturbing the demonstrations during the inference phase leads to a reduction of -1.3%/-2.2% on Arg-C F1 for both benchmarks. The reduction further increases to -1.8%/-3.1% when the perturbation is applied during both the training and inference phases. Despite experiencing a slight performance drop in the presence of incorrect demonstrations, DemoSG consistently outperforms the robust baseline UIE. The results underscore the effectiveness of demonstration-based learning, highlighting the influence of accurate demonstrations in helping model understand the extraction process.

2) The absence of demonstrations has a more significant impact than incorrect demonstrations. Specifically, when dropping demonstrations only during the inference phase, there is a reduction of -8.5%/-6.6% in Arg-C F1, resulting in DemoSG performing worse than UIE on ACE05-EN+. We consider that this phenomenon can be attributed to the Exposure Bias (Schmidt, 2019) between the training and inference phases, which also explains why DemoSG exhibits fewer improvements in data-limited settings. It is noteworthy that dropping demonstrations during both the training and inference phases leads to a recovery of +3.1%/+3.6% in Arg-C performance compared to *Test Drop*. These results suggest that reducing exposure bias could be an effective approach to enhance the robustness of demonstration-based learning. We leave further investigation of this topic for future studies.

## 5 Related Work

**Low-resource Event Extraction.** Most of the previous EE methods focus on the high-resource scenario (Nguyen et al., 2016; Yan et al., 2019; Cui et al., 2020; Du and Cardie, 2020a; Ramponi et al., 2020; Huang and Peng, 2021), which makes them hardly applied to new domains where the annotated data is limited. Thus, low-resource EE starts to attract attention recently. Lu et al. (2021) models EE as a unified sequence-to-structure generation, which shares the knowledge between the Event Detection and Argument Extraction subtasks. However, the gap between the downstream structure generation and pre-training natural language generation has not been considered. Based on Lu et al. (2021), Lu et al. (2022) proposes several pre-training strategies on large-scale datasets, which boosts the structure generation and improves the low-resource EE performance. However, large-scale pre-training needs a lot of computing resources and excess corpus. Hsu et al. (2022) incorporates additional information via manually designing a prompt for each event type, which contains the description of the event and the relation of roles. However, designing such delicate prompts still requires quite a few human efforts, and the EE performance is sensitive to the design. Recently, Lou et al. (2023) proposes unified token linking operations to improve the knowledge transfer ability. We leave the comparison to future studies since their code and data sampling details are not available.

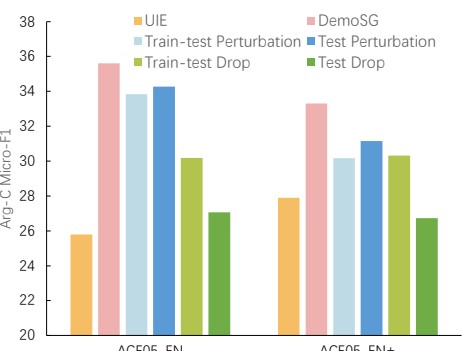

Figure 3: Experiment result of the robustness study under the 5-shot low-resource setting. DemoSG and its variants are based on *rich-role* in the experiments.

**Demonstration-based Learning.** The idea is originally motivated by the in-context learning ability of the GPT3 (Brown et al., 2020), where the model learns by only conditioning on the demonstration without tuning. However, in-context learning relies on the enormous size of PLMs. To make the demonstration effective for small PLMs, Lee et al. (2022) proposes a demonstration-based learning framework for NER, which treats demonstrations as additional contextual information for the sequence tagging model during training. Zhang et al. (2022) further studies the robustness of demonstration-based learning in the sequence tagging paradigm for NER. Our method differs from these studies from: 1) Our demonstration-enhaced paradigm is designed according to the specific characteristics of event extraction. 2) Our generative framework enhances the flexibility of demonstration-based learning, providing it with the parameter-agnostic domain-adaptation capability.

## 6 Conclusion

In this paper, we propose the Demonstration-enhanced Schema-guided Generation (DemoSG) model for low-resource event extraction, which benefits from two aspects: Firstly, we propose the demonstration-based learning paradigm for EE to fully use the annotated data, which transforms them into demonstrations to illustrate the extraction process and help the model learn effectively. Secondly, we formulate EE as a natural language generation task guided by schema-based prompts, thereby leveraging label semantics and promoting knowledge transfer in low-resource scenarios. Extensive experiments and analyses show that DemoSG significantly outperforms current methods in various low-resource and domain adaptation scenarios, and demonstrate the effectiveness of our method.

## Limitations

In this paper, we propose the DemoSG model to facilitate low-resource event extraction. To exploit the additional information of demonstrations and prompts, DemoSG generates records of each event type individually. Though achieving significant improvement in low-resource scenarios, the individual generation makes DemoEE predict relatively slower than methods that generate all records at once (Lu et al., 2021, 2022).

## Ethics Statement

The contributions of this paper are purely methodological, specifically introducing the Demonstration-enhanced Schema-guided Generation model (DemoSG) to enhance the performance of low-resource Event Extraction. Thus, this paper does not have any direct negative social impact.

## Acknowledgements

We sincerely thank all anonymous reviewers for their valuable comments and suggestions. This work was supported by Program for Youth Innovative Research Team of BUPT No. 2023QNTD02.

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

## A    Comparison of Model Parameters

We put the comparison of the PLMs and the amount of model parameters of DemoSG with other low-resource baselines in Table 6.

| Model | Backbone PLM | Params |
|-------|--------------|--------|
| OneIE | BERT-large | 308M |
| DEGREE | BART-large | 404M |
| Text2Event | T5-large | 780M |
| UIE | T5-large | 780M |
| DemoSG | BART-large | 404M |

Table 6: Comparison of model parameters.