# OpenReview forum: "DemoSG: Demonstration-enhanced Schema-guided Generation for Low-resource Event Extraction"
_EMNLP/2023/Conference — EMNLP 2023 Findings_

### Official Review · Reviewer_99nK · 2023-08-03

**Typos Grammar Style And Presentation Improvements:** Throughout the paper, change “domain …
**Soundness:** 4

**Excitement:**

4: Strong: This paper deepens the understanding of some phenomenon or lowers the barriers to an existing research direction.

**Paper Topic And Main Contributions:**

Summary: Building upon previous work in structured generation (Lu et al 2021, Lu et al 2022, Hsu et al 2022), this paper presents work in low-resource event extraction using NLG and schema-based prompts, where NLG involves large language seq2seq models with raw text input and structured representation output for events and event arguments. “Demonstration-enhanced” refers to sampling methods to select the best examples for in-context learning, while “schema-based prompts” are natural language event templates for each event type. By low-resource settings, the researchers mean how many examples the model sees (full supervision vs few-shot in-context learning). Their methods outperform other systems in low-resource scenarios, and the paper gives evidence for generative paradigms outperforming classification methods more generally. The main dataset used are variants of ACE2005, and code will be open source (some code available as attachment). Main contributions are experiments in structure generation and studying event extraction in low resource settings (data efficiency).

**Questions For The Authors:**

(Lines 304-310): I wonder what the effects of the negative sampling. Specifically, if there are 33 event types, then for a given mention one positive example is made and 32 negative examples which can be sampled from. How often does the model predict more than one possible event type for a given mention, and how do you select then from candidates?

(Line 320): The conversion of the generated text to a data object. How well does the rule-based deterministic algorithm work in practice? Is this just a literal_eval or something? How often does the model overgenerate for example, brackets or matching parentheses or whatever? This kind of evidence I think adds to the argument for the strengths of generative over classification models.

To make room for more error analysis and discussion, consider collapsing Section 2.2.1 and 2.2.3, which contain near identical content. Also, a part of the introduction (lines 41-72) is repeated nearly verbatim in the related work section (lines 579-606).

The researchers don’t take the opportunity to consider future directions for EE. A lot of work in EE (including this one) use data based on ACE2005. Is this satisfactory? What opportunities do LLMs make possible for future EE work?

To strengthen the related work section, you could also include citations for using LLMs for other kinds structure generation (IE-related as well as syntax, other semantic representations, etc).

**Reasons To Accept:**

Strengths:

The researchers show strong evidence of how to improve efficiency of the use of annotated data using demonstrations and schema-based prompts, thus they show strong evidence of how EE can be improved in low-resource settings. Seq2seq generation for event extraction, or structure generation, appears to work also in cross-domain EE with the additional signal of event participant labels.

To increase the strength of their argument, the researchers tested the hypothesis of cross-domain event meaning transfer, measuring how well does the model transfer knowledge between domains measuring the “parameter-agnostic domain (sic) adaption capability” of the model (section 3.3), testing how knowledge transfers across different events using schema-guided natural language generation (leveraging similarity between event types / entity types). See domain adaptation scenario (lines 171-181) and results reported (lines 466+). However, it is a missed opportunity to examine what factors are at play in increasing performance here (see below).

**Reasons To Reject:**

Example in Figure 1. I think this is a generally good figure, but it leaves a lot ambiguous to stand on its own. For example, the input text is shown with directed edges between events and their arguments. Is this graph information (including mention spans) part of input? Also, for each training input, you select how many demos? Does the color of the boxes around “select demo” examples and “schema” examples mean something (the red vs. the blue)? Is this to create a single positive example, or multiple? Does the training set look like this: [(input text + ?event mention + transport demo + transport schema), (input text + ?event mention + meet demo + meet schema), (input text + ?event mention + negative-event demo + negative-event schema), …]? And then at inference time, the model generates data for all event types? A lot of this could be of been made more clear in the figure or early in the paper.

(Line 72): You say that “EE is sensitive to prompt design”. Did you test any variants of prompts? Are there general best practices for EE prompt design that you found that helps explain some of the variance observed?

(section 2.2.2) Event demonstration construction processes are not clearly defined. What are the “best demonstrating characteristics” (lines 242-243)? How many demonstrations does the model select on average between the demonstration-oriented and the instance-oriented methods? How does this affect the decoding process, i.e. if a demo for a specific event type is not selected for a specific example that is indeed associated with that event type, how well does the model do in generalizing to “unseen” event types in the low-resource setting? What is the impact of this on performance of the “select demo” part of Figure 1? It would help to parse into natural language the functions in the included utils.py, for example, maybe as part of the Appendix.

Results section ignores the model’s weaknesses: there is not an error analysis, making it hard to understand better what still needs to be improved. How well does the model do in special cases? Does model performance reflect mostly good performance on the more frequent event types (say, Attack event?) How well does the model do for arguments associated with multiple arguments? Is all of the input limited to single sentences, or are there cases where arguments need to be extracted across larger contexts? More broadly, where does the model fail more predictably?

The ablation study is not airtight. The claim that adaptation was due to “leveraging the label semantic information of the event schema” needs some support to be more convincing. It is not clear how this was tested “removing the schema semantic information” (line 570): How is this done in practice, replacing semantic roles like “adjudicator” with an empty ARG label? Is it an apples-to-apples comparison? Also, does the schema-based prompts depend on the demonstration selection? If so, how does this affect the ablation study? And how to account for the big drop in performance when both schema and demos are removed (lines 573-577)?

An additional concern I have is the use (or overuse) of the same benchmark dataset (which has been around since 2005) which limits how much the event extraction field can be pushed towards new challenges, like reasoning about how participants change as the result of an event sequence, or selecting salient events, or building timelines of events, or summarization or something. What does the research suggest for the next iteration of experiments and the future of event understanding tasks in this day and age of LLMs?

**Reproducibility:**

3: Could reproduce the results with some difficulty. The settings of parameters are underspecified or subjectively determined; the training/evaluation data are not widely available.

**Reviewer Confidence:**

4: Quite sure. I tried to check the important points carefully. It's unlikely, though conceivable, that I missed something that should affect my ratings.

---

> ### Author Rebuttal · Authors · 2023-08-28
>
> We sincerely appreciate your acknowledgement for our work, meticulous review comments and valuable suggestions! We greatly concur with your concerns into the future of event extraction, and are willing to engage in discussions regarding the future of event extraction in the age of LLMs. Below is our detailed response to your questions and concerns:
>
> **Q1:** Towards questions about Figure 1: "Example in Figure 1. I think this is a generally good figure, but it leaves a lot ambiguous to stand on its own..."
> **A1:** Thanks for the feedback. The intention behind the design of Figure 1 is to provide an intuitive overview of the event extraction task and showcase our core innovation. Therefore, some details were simplified, and these details are elaborated upon more thoroughly in Figure 2. Here are our responses to your questions:
> - *"Is this graph information (including mention spans) part of input?"*
> The model takes only the training text as input. The arrows in the figure are included to provide a clearer illustration of the event composition in the example sentences, facilitating readers' understanding of our method.
> - *"Also, for each training input, you select how many demos?"*
> For each type of event, we choose only one demonstration. To the best of our knowledges, we are the first to introduce demonstration-based learning architecture for event extraction. The primary focus of our study is to explore the impact of introducing demonstrations on low-resource event extraction. We will leave the exploration of the optimal number of demonstrations for future work, and we will supplement the discussion in the camera-ready version to provide initial insights into how the number of demonstrations may affect performance.
> - *"Does the training set look like this:"* We didn’t include ‘mention’ as input. The actual structure of the training set is as follows: [(transport demo + input text + transport prompt), (meet demo + input text + meet prompt), (negative-event demo + input text + negative-event prompt), ...].We have provided formats of training examples in Figure2.
> - *"And then at inference time, the model generates data for all event types"* Yes. During the inference phase, extraction results are generated for each type of event.
> We will make according revisions to enhance the clarity of Figure 1 in our camera-ready version.
>
> **Q2:** Towards the misunderstanding about our description: "*EE is sensitive to prompt design*":
> **A2:** When we mention “EE is sensitive to prompt design”, we refer to DEGREE's incorporating supplementary information by manually designing a prompt for each event type, encompassing event descriptions and role relations. For instance, the additional description for the *Justice:Sue* event is *somebody was sued by some other in somewhere. The adjudication was judged by some adjudicator.* This fine-grained design makes the method highly sensitive to the event description, and this has been mentioned in their paper.
> Furthermore, practically, we believe that the natural language form of prompts, instead of the structured prompts used in Text2Event and UIE, is superior. This is because natural language prompts align better with the generation task guided by the model's pretraining, offering better compatibility with downstream tasks, bridging the gap between pretraining and fine-tuning, and promoting the transfer of pretrained knowledge. This design also allows our method to save a significant amount of further pretraining costs compared to UIE and Text2Event.
>
> **Q3:** Towards questions about demo construction process: "*What are the 'best demonstrating characteristics'...*":
> **A3:** The term 'best demonstration characteristics' refers to the perspective of selecting samples that possess superior demonstration capabilities. This selection is made from the standpoint of the demonstrations themselves. We have proposed two demonstration-oriented selection strategies: rich-role and rich-context.
> - Rich-role selects the sample involving the most event roles for each type of event as the demonstration for that event type. This strategy takes into consideration that training examples carrying more roles can better illustrate the event extraction process for that event type.
> - Similarly, rich-context selects the sample with the longest text as the demonstration, considering that longer contextual information may be helpful.
>
> Our approach only selects the most suitable sample to construct the demonstration, aiming to explore the benefits of using demonstrations for low-resource event extraction.
> The difference between our data-limited and few-shot experimental settings lies in the possibility of demonstration absence in the former (because we cannot guarantee that the training data covers all event types, during testing, the model may encounter event types and their demonstrations it has not seen before). However, the model can still generalize to examples without demonstrations through the event prompts, consistently improving DemoSG's overall performance over the baseline.
> Regarding the impact of demonstrations and the issue of their robustness, we conducted corresponding explorations in Section 3.5 Ablation Study as well as the Effectiveness and Robustness experiments in Appendix B. Our robustness experiments validate that mixing training samples with missing demonstrations during training can enhance the model's robustness and generalization ability in scenarios where demonstrations are missing.
>
> **Q4:** Towards the suggestion about weaknesses analysis:
> **A4:** In Appendix B, we delve into the robustness of DemoSG, showcasing areas where DemoSG has room for improvement, specifically cases where the model has demonstrations during training but lacks them during prediction. We also provide initial insights into how to address this weakness in future optimizations, namely by mitigating the exposure bias issue via introducing demonstrations during the training process. We will include further analysis of bad cases in the Case Study section of the camera-ready version.
>
> **Q5:** Towards questions about our ablation study:
> **A5:** We appreciate the insightful question. In our ablation study involving the use of schema semantic information, we removed event type descriptions and replaced the roles in the prompts with role indices. For example, "Origion is [mask]. Destination is [Mask]…" was replaced with "Role1 is [mask]. Role2 is [Mask]…", and the label semantics from the schema were removed. To ensure a fair ablation experiment, we simultaneously replaced both roles in the demonstrations and the prompts with role indices to maintain their consistency. Hence, we think it is an apples-to-apples comparison. Regarding the significant drop observed when simultaneously ablating schema and semantic information, we believe this phenomenon may be due to the fact that demonstrations and label semantics not only have a cumulative effect but also share some common influence. This may be a crucial factor for the success of generative models in low-resource event extraction tasks.
>
> **Q6:** Towards your concern "*An additional concern I have is the use (or overuse) of the same benchmark dataset...*" and the future of EE in the age of LLM:
> **A6:** As for your additional concern, This is a profound reflection. We agree with your perspective that the event extraction task has been extensively studied on benchmarks like ACE and ACE+, and it requires new and more challenging benchmarks. However, until such a innovative benchmark is proposed, we believe that future research in event extraction can focus on the following aspects:
> - **Low-resource event extraction:** Previous methods have mostly relied on large amounts of data. For the event extraction task that has high annotation costs, addressing event extraction under low-resource scenarios is challenging, which is also the motivation of our study.
> - **Multimodal Event Extraction:** With the increasing trend toward multimodal information, events are increasingly presented in the form of multimodal images and text. For example, news reports about certain events often come with accompanying images. Research on how to combine images to enhance performance in multimodal event extraction is a promising future research direction [1].
> - **Document-level event extraction:** Event extraction from long texts at the document level presents unique challenges (multi-event, argument-scattering [2]), and efficiently capturing cross-document event information is also a significant challenge, even for LLMs.
> - **More elegant integration of LLM capabilities:** LLMs have demonstrated their strong capabilities in multiple tasks. Directly modeling a task as a text generation and conducting instructional supervised fine-tuning on LLMs often yields improvements. However, fine-tuning LLMs consumes significant computational resources. Future research directions should explore more elegant ways to harness the advantages of LLMs. For example, in this paper, we propose the demonstration-based learning architecture for event extraction, bringing the in-context learning ability of LLMs to smaller PLMs (BART), enabling small models to understand event extraction tasks through demonstrations, and enhancing their low-resource event extraction capabilities. We believe that exploring methods that can elegantly combine the strengths of large models will be one of the future directions in the era of LLMs.
>
> **Q7:** Towards your question about effects of negative sampling: "*I wonder what the effects of the negative sampling...*":
> **A7:** Negative sampling is applied exclusively during the model training process to balance the ratio of positive and negative samples. A single sample often contains one or several types of events. For these included event types, the sample is considered a positive sample, while for the event types not included, it is considered a negative sample.
> Consequently, the total number of negative samples is quite large, and if without negative sampling, the model would tend to predict *no event occurred*, leading to a decrease in Recall scores. During prediction, there is no need for negative sampling.
>
>
> **Q8:** Towards the confusion about the conversion of the generated texts to event records:
> **A8:** During decoding, we match the defined templates: first, we segment multiple event records based on triggers, and then we locate the corresponding arguments based on the roles within each event record.
> We model event extraction as natural language generation rather than structured generation (as in UIE and Text2Event). This approach mitigates the gap between natural language generation tasks during PLM pretraining and downstream fine-tuning tasks. Therefore, our model can learn the output format even in low-resource scenarios without issues such as redundant parentheses.
>
>
> **Q9:** Towards the suggestion for improving the related work section:
> **A9:** Thank you for the suggestion. We will continue to monitor recent IE-related work that leverages LLMs and make appropriate additions to the related work section.
>
> **References**
> [1] Cross-media Structured Common Space for Multimedia Event Extraction
> [2] Document-level event extraction via heterogeneous graph-based interaction model with a tracker

---

### Official Review · Reviewer_7B7C · 2023-08-04

**Typos Grammar Style And Presentation Improvements:** N/A
**Soundness:** 3

**Excitement:**

3: Ambivalent: It has merits (e.g., it reports state-of-the-art results, the idea is nice), but there are key weaknesses (e.g., it describes incremental work), and it can significantly benefit from another round of revision. However, I won't object to accepting it if my co-reviewers champion it.

**Missing References:**

N/A

**Paper Topic And Main Contributions:**

This paper introduces the DemoSG model for low-resource Event Extraction (EE). It employs demonstration-based learning and schema-guided generation to enhance performance in low-resource scenarios. More details on methodology and results are needed, but the approach holds promise for improving EE in resource-scarce contexts.

The paper acknowledges the drawback related to computation cost; however, it overlooks a potential limitation concerning the input length.

**Questions For The Authors:**

1. Could you provide more insight into the training time and inference time of the model, including details beyond what is presented in Table 6? Additionally, could you consider including metrics like GMACs or GFLOPs for the inference process of the proposed method to further enhance the analysis?

2. The concept of demonstrations generating records for each event type individually (as mentioned in line 304) is intriguing. However, it's not entirely clear how this approach aids unrelated event types. Has there been any assessment, either qualitative or quantitative, of the retrieval quality of demonstrations in these scenarios?

3. Considering the limitations of the pre-trained language model (BART), how was the optimal number of demonstrations determined? Could you elaborate on the selection process for this number? Also, is there a possibility that the length of the target instance might exceed the model's input limit?

4. The similarity between ACE05 and ACE05+ in Table 5 and Figure 3 raises questions about the selection of these two variants as benchmarks. Could you provide more clarity on why these specific benchmarks were chosen over other well-known event extraction benchmarks, such as wiki-event?

5. The application of rich-role and rich-context strategies (lines 249 - 252) could benefit from an illustrative example for better understanding. Specifically, how are these strategies implemented? Furthermore, considering ACE05's instances with multiple event records of varying types, could you explain how these selection strategies are adapted to handle such complexities?

**Reasons To Accept:**

- The paper has a well-organized structure that enhances its comprehensibility.

- The noticeable enhancements in results achieved by the proposed approach contribute to the paper's strength.

**Reasons To Reject:**

- The lack of experiments on diverse datasets weakens the generalizability of the findings.

- The computational complexity of the proposed method could potentially hinder its practical applicability.

**Reproducibility:**

4: Could mostly reproduce the results, but there may be some variation because of sample variance or minor variations in their interpretation of the protocol or method.

**Reviewer Confidence:**

4: Quite sure. I tried to check the important points carefully. It's unlikely, though conceivable, that I missed something that should affect my ratings.

---

> ### Author Rebuttal · Authors · 2023-08-28
>
> Thanks greatly for your suggestions, especially for the recommendation of supplementing experiments on other benchmarks! The following presents our supplementary experimental results, and our detailed response to your concerns and questions:
>
> **Q1:** *The lack of experiments on diverse datasets weakens the generalizability of the findings. The similarity between ACE05 and ACE05+ in Table 5 and Figure 3 raises questions about the selection of these two variants as benchmarks. Could you provide more clarity on why these specific benchmarks were chosen over other well-known event extraction benchmarks, such as wiki-event?*
> **A1:** We fully understand your concerns and share the same perspective that diverse datasets could provide a more comprehensive validation of our method's effectiveness. ACE05 and ACE05+ serve as the two predominant baselines in event extraction, as highlighted in recent literature [1-3] with their distinguishing attributes. In addition to ACE05 and ACE05+, We initially planned to extend our experiments to the ERE-EN [4] dataset for a more comprehensive study. However, acquiring the license for ERE-EN took some time. During the review period, we were able to supplement our work with experiments on the ERE-EN dataset.
> The experiments on ERE-EN also present the same trend with ACE05 and ACE05+, showing that DemoSG has a superior low-resource EE ability over baseline methods. Here, we present the salient results of low-resource experiments in the following tables. Comparing DemoSG against UIE, we can observe a 1.6% increase in Arg-C F1 on AVG-R, and a 4.8% increase on AVG-S, which also proves the better low-resource EE ability of out method. The full results on the ERE-EN benchmark will be provided in the camera-ready version.
> **Trig-C:**
> |     |Model   |  2-shot|  5-shot|  10-shot | AVE-S     | 2%|  5%|  10% | AVE-R  |
> | ----------- | ------------ | ------------ | ------------ | ------------ | ------------ | ------------ | ------------ | ------------ | -------- |
> |     |  UIE       |  32.2|  36.5  |  39.5  |  36.1      |  **33.1**  |  43.9  |  49.3  |  42.1  |
> | |  DemoSG-Role  |  32.6  |  43.2  |  **46.5**  |  40.8      |  32.3  |  **44.7**  |  **49.9**  |  **42.3**      |
> | |  DemoSG-Context  |  **36.0**  |  **45.0**  |  46.1  |  **42.4**      |  28.5  |  40.2  |  48.0  |  38.9      |
> | |  DemoSG-Similar  |  35.7  |  42.3  |  44.6  |  40.9      |  32.6  |  43.8  |  47.2  |  41.2      |
>
> **Arg-C:**
> |   |Model   |  2-shot|  5-shot|  10-shot | AVE-S     | 2%|  5%|  10% | AVE-R  |
> | ----------- | ------------ | ------------ | ------------ | ------------ | ------------ | ------------ | ------------ | ------------ | -------- |
> |      |  UIE       |  20.1  |  25.9  |  30.5  |  25.5      |  **19.1** |  29.9 |  35.8 |  28.3  |
> | |  DemoSG-Role  |  18.5  |  **30.9**  |  **38.4**  |  29.3      |  18.2  |  **33.2**  |  38.3  |  **29.9**      |
> | |  DemoSG-Context  |  22.3  |  **30.9**  |  37.8 |  **30.3**      |  17.3  |  31.9  |  38.9  |  29.4      |
> | |  DemoSG-Similar  |  **22.9** |  30.8  |  36.0  |  29.9      |  18.4  |  32.2  |  **39.2** |  **29.9**      |
>
>
> **Q2:** *The computational complexity of the proposed method could potentially hinder its practical applicability. Could you provide more insight into the training time and inference time of the model, including details beyond what is presented in Table 6? Additionally, could you consider including metrics like GMACs or GFLOPs for the inference process of the proposed method to further enhance the analysis?*
> **A2:** Though mentioned in our Limitation section that utilizing separate decoding processes may marginally increase the inference time, the resulting time delay is acceptable. Our training and inference times align closely with those of the recent DEGREE baseline. In comparison to structured generation techniques (UIE and Text2Event) which have extra costs for additional pre-training, DemoSG can even save the entire pretraining costs while introducing an acceptable increase in inference time due to the generation of natural style. Please refer to the following table for a comparison of training and prediction times on ACE05. Besides, thanks for the suggestion of introducing GMACs and GFLOPs metrics. Due to the tight timeframe for author responses, we will include the analysis of GFLOPS in the camera-ready version.
> | Model               | Training time (s/example) | Inference time (s/example) |
> |----------------------|--------|----------|
> | Text2Event | 0.4085 | 0.0557   |
> | UIE | 0.3175   | 0.0173   |
> | DEGREE | 0.3012   | 0.4608   |
> | DemoSG | 0.3699   | 0.4800   |
>
> \* The above results are obtained under consistent experimental environments, including identical batch sizes and computational resources, etc.
>
> **Q3:** *Considering the limitations of the pre-trained language model (BART), how was the optimal number of demonstrations determined? Could you elaborate on the selection process for this number? Also, is there a possibility that the length of the target instance might exceed the model's input limit?*
> **A3:** We choose only one sample to construct the demonstration. As the pioneering introduction of the demonstration-based learning for EE, our primary exploration focuses on the impact of incorporating demonstrations to enhance data utilization for low-resource event extraction. Hence, the investigation into the optimal number of demonstrations will be deferred to future research endeavors.
> Throughout our practical experimentation, we observed that selecting a single demonstration would not breach the model's input limitations. For employing more demonstrations or dealing with document-level input texts in the future studies, we believe that using PLMs with extrapolatable positional embeddings can offer a resolution.
>
> **Q4:** *The application of rich-role and rich-context strategies (lines 249 - 252) could benefit from an illustrative example for better understanding. Specifically, how are these strategies implemented? Furthermore, considering ACE05's instances with multiple event records of varying types, could you explain how these selection strategies are adapted to handle such complexities?*
> **A4:** When selecting demonstrations for each type of event, we begin by searching the training dataset for samples containing records of these specific event types. Records pertaining to other event types are filtered out from consideration, yielding a set of candidate samples. Subsequently, in accordance with the criteria of demonstration selection (rich-role, rich-context, similarity), samples are chosen from the candidate set. These selected samples are then structured into demonstration sequences using our proposed unified serialization format. In the scenario where a sample encompasses multiple records belonging to different event types, its demonstration for each event type exclusively takes into account the records associated with that particular event type. This design serves to alleviate the issue of interference among event types.
>
> **Q5:** *The concept of demonstrations generating records for each event type individually (as mentioned in line 304) is intriguing. However, it's not entirely clear how this approach aids unrelated event types. Has there been any assessment, either qualitative or quantitative, of the retrieval quality of demonstrations in these scenarios?*
> **A5:** For each event type, we solely consider samples that contain records of that particular event type for constructing demonstrations, and eliminate any records of other event types. In this way, the selected demonstrations inherently match the corresponding event types. This approach effectively resolves issues related to unrelated event types and potential interference among diverse event types.
>
> **References**
> [1] A Joint Neural Model for Information Extraction with Global Features
> [2] Text2Event: Controllable Sequence-to-Structure Generation for End-to-end Event Extraction
> [3] Unified Structure Generation for Universal Information Extraction
> [4] From light to rich ERE: annotation of entities, relations, and events

---

### Official Review · Reviewer_XB14 · 2023-08-05

**Soundness:** 4

**Excitement:**

4: Strong: This paper deepens the understanding of some phenomenon or lowers the barriers to an existing research direction.

**Paper Topic And Main Contributions:**

SUMMARY. The paper proposes and evaluates a new event extraction method tailored for low-resource scenarios named "Demonstration-enhanced Schema-guided Generagion (DemoSG)". To that purpose, DemoSG relies (i) on the capabilities of a pre-trained (fine-tuned) encoder-decoder LM (BART), (ii) on the inclusion of a 'demonstration' (i.e., example text + verbalized extracted event) of the event type to extract among the LM encoder inputs, and (iii) on leveraging the information in event type and role labels by including them in a 'schema-based prompt'. The decoder output is then matched (via string-matching) to trigger and argument mentions in the input text. DemoSG is evaluated in both in-domain (both low- and high-resource) and domain adaptation settings based on ACE05-EN and ACE05-EN+ datasets, where it is compared to several state-of-the-art systems for both low- and high-resource event extraction, exhibiting overall performance improvements or anyhow competitiveness in the considered scenarios.

CONTRIBUTIONS:

* C1. Proposal of the DemoSG model, distinguished in particular by its 'unified representation template' to verbalize demonstrations and prompt, and the selection strategies to choose the demonstration events.

* C2. Experimental evaluation addressing different low/high-resource and in-domain/domain-adaptation settings, and including additional ablation and robustness (w.r.t. employed demonstration, in appendix) tests for DemoSG.

* C3. Improvements over state-of-the-art methods in many of considered experimental settings.

POST-REBUTTAL UPDATE. I thank the authors for the clarifications and the additional data in their rebuttal. After reading and considering it (esp. the supplied running times) and in the light of other reviews as well, I increased the *excitement* score from 3 to 4 to better reflect my leaning positively towards this submission.

**Questions For The Authors:**

* A. In relation to the need to iterate over all event types during extraction (also reported among limitations), what's the execution time of DemoSG compared to other systems? Can DemoSG deal with large event ontologies with many event types?

**Reasons To Accept:**

* S1. Solid performance in all considered settings, with substantial improvements in some settings and especially in relation to argument extraction.

* S2. Reproducibility facilitated by adequate description of the approach in the paper and by code provided as part of supplemental material.

* S3. Overall well written paper.

**Reasons To Reject:**

* A. In relation to the need to iterate over all event types during extraction (also reported among limitations), what's the execution time of DemoSG compared to other systems? Can DemoSG deal with large event ontologies with many event types?

* B. The ideas underlying the proposed DemoSG approach build on well known techniques (e.g., 'demonstration' w.r.t. in-context learning) and the contribution feel overall incremental in its nature.

**Reproducibility:**

5: Could easily reproduce the results.

**Reviewer Confidence:**

2: Willing to defend my evaluation, but it is fairly likely that I missed some details, didn't understand some central points, or can't be sure about the novelty of the work.

**Typos Grammar Style And Presentation Improvements:**

* T1.  [§2.2.1] "Multiple arguments correspond to" -> "that correspond to" / "corresponding to"
* T2.  [§2.3] "that is closet" -> "closest"
* T3.  [Eq. 3] is something missing here to differentiate between samples from D_P and D_N?
* T4.  [Eq. 4] p(Y|...) and p(y_i|...) should be conditioned also on input sentence X and parameters \theta

---

> ### Author Rebuttal · Authors · 2023-08-28
>
> Thank you sincerely for recognizing our work and providing valuable suggestions! Below is our detailed response to your concerns:
>
> **Q1:** *In relation to the need to iterate over all event types during extraction (also reported among limitations), what's the execution time of DemoSG compared to other systems? Can DemoSG deal with large event ontologies with many event types?*
> **A1:** Though mentioned in our Limitation section that utilizing separate decoding processes may marginally increase the inference time, the resulting time delay is acceptable. Please refer to the following table for a comparison of training and prediction times on ACE-05. Our training and inference times align closely with those of the recent DEGREE baseline. In comparison to structured generation techniques (UIE and Text2Event) which have extra costs for additional pre-training, DemoSG can even save the entire pretraining costs while introducing an acceptable increase in inference time due to the generation of natural style.
> Moreover, DemoSG could deal with large event ontologies with many event types. Note that ACE-05 associates with 33 types of events, which can already be considered as having a biggish event ontology. For situations with even larger event ontologies, DemoSG can also be tailored accordingly via a quick event type recall to identify potential event types, thereby saving inference time significantly.
> | Model               | Training time (s/example) | Inference time (s/example) |
> |----------------------|--------|----------|
> | Text2Event | 0.4085 | 0.0557   |
> | UIE | 0.3175   | 0.0173   |
> | DEGREE | 0.3012   | 0.4608   |
> | DemoSG | 0.3699   | 0.4800   |
>
> \* The above results are obtained under consistent experimental environments, including identical batch sizes and computational resources, etc.
>
> **Q2:** *The ideas underlying the proposed DemoSG approach build on well known techniques (e.g., 'demonstration' w.r.t. in-context learning) and the contribution feel overall incremental in its nature.*
> **A2:** While our approach draws inspiration from In-context Learning, it's important to emphasize the significant differences. In-context Learning relies on the “emergent” capabilities of large language models with a vast number of parameters (e.g., GPT-3.5 with over 100 billion parameters) and extensive pretraining. It does not necessarily require the inclusion of Demonstrations during training, and the choice of Demonstrations can be relatively arbitrary. However, this approach is not suitable for smaller PLM models that prioritize efficient inference speeds, for their weakness in parameter scale. Our goal is to enable Demonstrations to be effective even on models with relatively fewer parameters (similar to models like BART), and efficiently address low-resource event extraction. Hence, in our design of DemoSG, we take various factors into consideration to ensure its demonstration-based learning compatibility:
> **1. Introducing Demonstrations during Training:** Different from the In-context Learning of LLMs，we have incorporated demonstrations during the training process, enabling the BART model to acquire the ability of understanding the provided demonstrations.
> **2. Designing Demonstrations Selection strategies:** We have meticulously tailored the methods of selecting Demonstrations to ensure the inclusion of high-quality examples.
> **3. Modeling EE as Natural Language Generation:** By transforming the extraction task into a generation task, we unify the forms of demonstrations, prompts, and model outputs, fostering their intrinsic correlations and facilitating the transfer of knowledge across different event types.
>
> **Q3:** Regarding the questions mentioned in "*Typos Grammar Style And Presentation Improvements*":
> **A3:** We greatly appreciate your corrections! We will rectify the relevant problems in the camera-ready version. Regarding the question in T3, we think there are no missing elements in the formula. Because both positive and negative samples are treated uniformly as they undergo the same text generation process.

---

### Meta-Review · Area_Chair_mfUx · 2023-09-07

**Recommendation:** 4

**Metareview:**

The paper introduces DemoSG to handle event extraction low-resource scenarios. DemoSG leverages BART and incorporates a 'demonstration' of the event type to extract, along with schema-based prompts that include event type and role labels. The BART decoder output is matched to trigger and argument mentions in the input text. The evaluation covers in-domain and domain adaptation settings using ACE05-EN and ACE05-EN+ datasets, demonstrating competitive or improved performance compared to state-of-the-art systems for low- and high-resource event extraction.



The reviewers raised considerable concerns but the reviewers could convince the reviewers and address many of their concerns.  Additionally, the reviewers also agree that the contribution is exciting and the results are reproducible.

---

### Decision · Program_Chairs · 2023-10-07

**Decision:**

Accept-Findings

**Comment:**

The paper introduces DemoSG to handle event extraction low-resource scenarios. DemoSG leverages BART and incorporates a 'demonstration' of the event type to extract, along with schema-based prompts that include event type and role labels. The BART decoder output is matched to trigger and argument mentions in the input text. The evaluation covers in-domain and domain adaptation settings using ACE05-EN and ACE05-EN+ datasets, demonstrating competitive or improved performance compared to state-of-the-art systems for low- and high-resource event extraction.



The reviewers raised considerable concerns but the reviewers could convince the reviewers and address many of their concerns.  Additionally, the reviewers also agree that the contribution is exciting and the results are reproducible.